# Modelling Foreign Exchange Interventions under Rayleigh Process: Applications to Swiss Franc Exchange Rate Dynamics

**DOI:** 10.3390/e24070888

**Published:** 2022-06-28

**Authors:** Cho-Hoi Hui, Chi-Fai Lo, Chi-Hei Liu

**Affiliations:** 1Hong Kong Monetary Authority, 55/F, Two International Finance Centre, 8, Finance Street, Central, Hong Kong, China; 2Institute of Theoretical Physics and Department of Physics, The Chinese University of Hong Kong, Shatin, N.T., Hong Kong, China; cflo@phy.cuhk.edu.hk (C.-F.L.); christopherhtyc@gmail.com (C.-H.L.)

**Keywords:** exchange rate dynamics, Rayleigh process, quasi-bounded process, interventions

## Abstract

This paper models the foreign exchange intervention policy following the Rayleigh process derived from the standard flexible-price monetary framework. The exchange rate dynamics associated with the interventions are more sensitive to the change in the economic fundamental when a currency’s money supply is ample and its appreciation expectation cannot be offset by lower interest rates that have fallen to the zero lower bound, suggesting that more intensive interventions are required to counteract currency appreciation pressure and resulting in foreign reserve accumulation. The empirical results using market data during January 2015–February 2020 demonstrate that the model can describe the dynamics of the Swiss franc exchange rate. The accumulation of foreign reserves through interventions is negatively co-integrated with the exchange rate volatility and the value of the mean level of the Swiss franc exchange rate in the dynamics, to some extent indicating a reasonably high degree of effectiveness of the Swiss National Bank’s interventions. The transition between the target-zone and floating-rate regimes in 2015 caused changes in the level of exchange rate volatility but not its dynamical structure, suggesting that transitions between the floating-rate and target-zone regimes do not seem to have material consequence in this regard.

## 1. Introduction

After the global financial crisis, the interest rates in some developed countries, including Japan, the euro area, Switzerland, the US, and the UK, were low and fell close to zero or even negative. When nominal interest rates of the home and foreign countries are at zero, home currency assets will be more attractive to foreigners if there is an expected future appreciation of the home currency that cannot be offset by lower domestic interest rates at the zero (or negative) lower bound (ZLB). The binding ZLB becomes a constraint for a consistent adjustment between interest rate parity and currency appreciation expectation. To relieve their currency appreciation under this constraint in the face of turmoil in financial markets and capital inflows, foreign exchange interventions were undertaken by the governments/central banks of Japan, Switzerland, Israel, and New Zealand, whose currencies are under a floating-rate regime. Their interventions result in foreign reserves accumulated by the central banks. While the results of empirical studies on the efficiency of the interventions are mixed, they empirically find that this policy can successfully weaken the exchange rates [1]. These interventions by monetary authorities are, however, costly [2].

Even under the floating-rate regime, some central banks intervene in the foreign exchange market [3,4,5,6,7,8,9,10,11]. The objective of the interventions is to stabilise or manage exchange rate dynamics in terms of their levels and volatility, at least in the short-run. For example, the Japanese government intervened in 2010–2011, in coordination with other G-7 countries, after the yen appreciated to a record high against the US dollar, breaking through 80 yen for the first time since 1995. Empirical results show that high frequency intervention stabilises the exchange rate by reducing exchange rate volatility and that low frequency intervention is more effective for changing exchange rate levels [8]. Authorities in both developed and emerging market countries operate their foreign exchange interventions according to their exchange rate/monetary policies or on a necessary basis. Most currency interventions were coordinated among multiple government agencies to enhance their effectiveness [3,4]. Such interventions for currencies under the floating-rate regime to some extent exerted a stabilising influence on their exchange rate levels and volatility [11].

In the case of Switzerland, as the European sovereign debt crisis deepened from 2010, investors were prompted to seek safe havens because of the weakening euro (EUR). As a result, the Swiss franc (CHF) came under tremendous upward pressure. Despite a zero or negative nominal interest rate, Switzerland has experienced a large increase in private capital inflows since 2010 that was accompanied by an equally large increase in the foreign reserves held by the Swiss National Bank (SNB) due to its interventions, which have prevented an appreciation of the Swiss franc. In response to the upward pressure on the Swiss franc, the SNB on 6 September 2011 put a ceiling on the value of the Swiss franc at 1.2 per euro (1/1.2 EUR/CHF) and vowed to enforce this limit “with utmost determination” and to buy foreign currencies “in unlimited quantities” [12]. This measure effectively brought its exchange rate system from a floating-rate regime to a one-sided target-zone regime. However, there was the divergence of monetary policies in the US and the euro area, which lead to a strengthening of the US dollar and a weaker euro. Maintaining the floor would have required increasingly large interventions and weakened the CHF against the dollar and other currencies (due to euro weakness). With continuous weakness in the euro area economy, the SNB surprised the market by abandoning its exchange rate limit on 15 January 2015, sending the Swiss franc back to the floating-rate regime and up by 15% against the euro in a few days. While the SNB was expected to gain some degree of independence for its monetary policy, given the interest rate at the ZLB, the SNB intervened more heavily in the foreign exchange market to prevent an appreciation of the Swiss franc, and accumulated more reserves after abandoning the exchange rate ceiling (see Figure 1). Because of the impact of the coronavirus pandemic in early 2020, the SBN counteracted the increased upward pressure on the Swiss franc and decided to scale up its foreign exchange market inventions to shield the Swiss economy. This is in line with the role of the Swiss franc being treated as a safe haven currency by market participants during periods of market turmoil in particular [13,14,15,16].

To study the exchange rate dynamics with a central bank’s interventions as in the case of Switzerland, we extend the quasi-bounded target-zone model [17,18] based on the standard flexible-price monetary framework to develop an exchange rate model by which we can derive the exchange rate dynamics incorporated with interventions in a floating-rate regime. The model is based on the existence of a money demand function, the purchasing power parity and uncovered interest rate parity (UIP). UIP is an arbitrage relationship that links the interest rate differential (risk-free rates) to the expected currency appreciation. To incorporate interventions into the model, the economic fundamental dynamics follow a mean-reverting process in which a restoring force moves the fundamental towards a mean level and its magnitude is proportional to the deviation from the mean. One driving force behind the mean reversion could be attributed to the strategy of “leaning against the wind” adopted by a monetary authority. It represents an error-correction action of interventions taken by the monetary authority to pull the exchange rate back to its long-run equilibrium whenever the currency appreciates too much or its appreciation pressure is too large. The corresponding mean-reverting fundamental shock is asymmetric given that interventions are one-sided events at the strong side that are inherently asymmetric.

To solve the exchange rate equation in the model, a smooth-pasting boundary condition is imposed at a boundary that is set at a very substantial appreciation level. Subject to this boundary condition, we show that the fundamental dynamics following the Rayleigh process is uniquely determined for the exchange rate equation [19]. The Rayleigh process is considered to be a generalised Bessel process, or a Bessel process with a linear drift or radial Ornstein–Uhlenbeck process related with the description of optical field, as it generalises the radial projection of a *d*-dimensional Ornstein-Uhlenbeck equation [20,21]. Such processes applied for the Brownian motion assume that there is an equilibrium that is stable. It is noted that when there are shocks such as inflation surprises in an economy, its currency’s exchange rate may not be in equilibrium or the equilibrium is not stable due to speculative trading [22,23]. The Rayleigh process is applied to the Gompertz-type models for a birth-death process [24], tumor growth [25], energy consumptions [26], and option-pricing theory in finance [27]. Some quantities in these examples are conserved, while exchange rates could have chaotic fluctuations [28].

The distribution generated from the Rayleigh process is demonstrated to be important to information and communication theory in relation to sampling and censoring [29,30,31]. With regard to this importance and the desire to give greater flexibility to this distribution, several researchers have developed extensive extensions to the Rayleigh distribution [32,33,34]. One of the methods of deriving distributions is the optimisation of a measure of entropy, such that densities of univariate and multivariate distributions can be derived including Maxwell–Boltzmann and Rayleigh densities [35]. This paper shows that the foreign exchange intervention policy following the Rayleigh process gives a probability-leakage condition for the exchange rate breaching a boundary, which forms a measure of entropy or a measure of uncertainty about degree of effectiveness of interventions.

The Rayleigh process exhibits an asymmetric mean-reverting property, more vigorous interventions are conducted by a monetary authority when the exchange rate moves closer to the boundary, i.e., more substantial appreciation of the currency, in order to keep the exchange rate away from the boundary. This is consistent with the intervention policy to prevent an appreciation of the currency, where there is no predetermined exchange rate level for interventions under a floating-rate regime. Such interventions suggest that exchange rates could move within a wide unannounced band or below a boundary as the results of interventions. The boundary is set by assuming that a monetary authority’s interventions benchmark a moving average of the current and past exchange rates rather than the exchange rate’s current level. It should be noted that, as long as the boundary represents a significant appreciation, the choice of its level does not affect the process of the exchange rate dynamics.

When the domestic interest rate is at the ZLB, the currency appreciation expectation and capital inflows cannot be offset by lowering the interest rate. Therefore, the interest rate differential cannot be widened further. Due to such constraint, the monetary authority needs to intervene in the foreign exchange market and counteract the upward pressure on its currency, resulting in foreign reserve accumulation. The empirical results using market data during January 2015–February 2020 demonstrate that the accumulation of foreign reserves due to the SNB’s interventions has a negative relationship with the exchange rate volatility and the strength of the mean level of the Swiss franc exchange rate in the mean-reverting square-root dynamics derived from the model, to some extent indicating a reasonably high degree of effectiveness of the SNB’s interventions at the ZLB. Studies on the open economy dimension of the ZLB are in recent research [36,37,38,39,40,41,42,43,44].

The paper is organised as follows. We present the exchange rate model incorporated with interventions in the fundamental dynamics in the following Section. The corresponding exchange rate dynamics, interest rate differential, and exchange rate distribution are shown in Section 3. The calibrations of the exchange rate dynamics based on the Swiss franc exchange rate are presented in Section 3.4. The relationship between the exchange rate dynamics and the SNB’s foreign reserves is studied empirically and discussed in Section 4. The final section is the conclusion.

## 2. Methods

### 2.1. Basic Exchange Rate Model and Boundary Conditions

To model a monetary authority’s intervention policy and the associated exchange rate dynamics, we use a standard law of motion for a flexible-price exchange-rate model that is based on the existence of a money demand function, the purchasing power parity, and the uncovered interest rate parity (UIP). This framework is used in the standard exchange rate target-zone model [45]. The log exchange rate at time *t* follows the equation:(1)$$s\left(t\right)=m+\nu \left(t\right)+\alpha \frac{E\left[ds\left(t\right)\right]}{dt}$$
where−∞<s≤0, *m* is the logarithm of the money supply, *α* is the absolute value of semi-elasticity of the exchange rate with respect to its expected rate of change, and *E* the expectation operator. The last term captures the expected exchange rate change under UIP and the time-*t* information set. The effect of the domestic interest rate falling to the ZLB on the exchange rate and exchange rate expectation is thus incorporated through this term. The stochastic variable ν(t) measures all factors other than the money supply and expected exchange rate. A monetary authority may intervene to influence exchange rate by adjusting the fundamental ν(t). Specifically, the authority can manage ν(t) to ensure that the exchange rate *s* will not breach a boundary of *s* = 0. For example, the authority could intervene with a monetary expansion to prevent the exchange rate from breaching *s* = 0 and interferes with the motion of ν when *s* is expected or has the tendency going towards *s* = 0. This suggests that it is not necessary for the authority to conduct interventions only at the boundary of *s* = 0. This model assumes a continuous-time representation, which has its limitation as the exchange rate is not a conserved quantity.

The “fundamental” (*ν*) is assumed to follow a stochastic process with a drift $\mu_{\nu }$, which is a function of *ν*:(2)$$d\nu =\mu_{\nu }dt+\sigma_{\nu }dZ$$
where $\sigma_{\nu }\;$is the random shock and dZ is a Wiener process with E[dZ]=0 and $E\left[dZ^{2}\right]=dt$. We apply Ito’s lemma to Equations (1) and (2), and have
(3)$$\frac{1}{2}\alpha \sigma_{\nu }^{2}\frac{d^{2}s}{d\nu^{2}}+\alpha \mu_{\nu }\frac{ds}{d\nu }-s=-\nu -m$$
which is a second-order linear ordinary differential equation. To solve Equation (3) with s∈(−∞, 0], we specify the following boundary conditions at the fundamental of ν=0:(4)s(ν=0)=0;
(5)$${\frac{ds\left(\nu \right)}{d\nu }|}_{\nu =0}=0$$
where the former condition ensures a proper normalisation of the exchange rate and the latter is the smooth-pasting boundary condition at ν=0, suggesting an optimal boundary condition for the process with no foreseeable jump in the exchange rate and no arbitrage condition [46]. Under a floating-rate regime, the boundary is moving and not fixed. If the condition does not hold, the exchange rate could jump across the boundary. Whatever the intervention policy may be, this condition provides sufficient boundary information to solve Equation (3). To ensure this boundary condition valid, the monetary authority will conduct infinitesimal interventions at some level of *ν* < 0, such that the exchange rate will not breach *s* = 0.

The boundary is defined as a monetary authority’s tolerance limit for a substantial revaluation of its currency or a distribution of the exchange rate’s statistics. Historical exchange rates can be used as a guide to set a monetary authority’s tolerance level of the exchange rate [47]. The historical trend of the exchange rate can be measured by a moving average $S_{A}\left(t\right)$ of the current and past exchange rate *S_t_*, which is often used to smoothen the price trends and optimise profits for certain trading strategies employing price fluctuations around those averages [48]. This shifts the reference for intervention from the level of the exchange rate at each instant to the behaviour of the exchange rate over a time interval. The moving average is scaled by a parameter $\eta_{U}$, such that $\eta_{U}S_{A}\left(t\right)$ forms a boundary for the exchange rate movement. The parameter $\eta_{U}$ reveals how far a monetary authority tolerates an appreciation of its currency or how much they expect the maximum or extreme upside of the currency.

Given the appreciation pressure is a one-sided process, the normalised log exchange rate *s* is specified as:(6)$$s=-\ln\left[\frac{S\left(t\right)}{\eta_{U}S_{A}\left(t\right)}\right]$$
where the denominator $\eta_{U}S_{A}\left(t\right)$ the upper boundary in the original exchange rate *S*. This qualifies how big is a change in the exchange rate *S* (the original exchange rate measure) under a floating-rate regime.

### 2.2. Fundamental Dynamics Incorporated with Intervention Policy

The drift $\mu_{\nu }$ in Equation (2) of the fundamental dynamics represents the behaviour of a monetary authority that conducts interventions in the foreign exchange market. The fundamental v, which is uniquely determined under the boundary conditions in Equations (4) and (5) for the exchange rate equation Equation (3), follows the Rayleigh process [19]:(7)$$d\nu =\left(\frac{A_{-1}}{\nu }+A_{1}\nu \right)dt+\sigma_{\nu }dZ$$
where *A*_1_ < 0, *A*_−1_ > 0, −∞<ν≤0. Using this fundamental dynamics, the associated exchange rate dynamics and interest rate differentials are derived from their target-zone model and can describe the market data for the Hong Kong dollar against the US dollar in a target zone and the Swiss franc’s one-sided target-zone regime during September 2011–January 2015, respectively [17,18].

Monetary authorities intervene in the foreign exchange market not only under the target-zone regime. In reality, their interventions under the managed or floating-rate regimes often occur in the market [10,11]. Therefore, the unique fundamental dynamics under the Rayleigh process is readily applicable to the floating-rate regimes where monetary authorities may intervene at certain levels of exchange rates. The asymmetric feature of the Rayleigh process is similar to asymmetric country-specific and global shocks in the context of contributions to violations of UIP [49] and exchange rate option [50,51]. In the case of the Swiss franc being a safe haven currency whose value increases with global risk aversion, several studies [13,52,53] document the role of global risk factors for the value of the Swiss franc.

The drift term in the Rayleigh process exhibits a mean-reverting property for the fundamental dynamics. When |v| is small (approaching to the zero boundary), the term$\;A_{-1}/\nu \;$in Equation (7) will push *v* away from zero. Such dynamics represent the fact that the monetary authority will conduct interventions at some level of *v* in order to move the exchange rate away from the level of *s* = 0 towards some targets (or mean levels), which are shown to be time-varying in the next section. In the context of capital inflows under appreciation pressure, the monetary authority will allow monetary expansion by buying foreign currencies and selling its domestic currency in the market. When *v* is far away from the origin, the effect of the term$\;A_{-1}/\nu \;$weakens, suggesting no need for interventions. Conversely, the term $A_{1}\nu$ will take place to push *v* back towards the origin, indicating that the monetary authority may intervene by monetary contraction to revert the exchange rate movement if the currency is considered to be too weak.

The two terms ($A_{-1}/\nu \;$and $A_{1}\nu )\;$in the Rayleigh process determine the mean-reverting process of the fundamental *v*, which is interpreted as a monetary shock, reflecting an error-correction policy on the part of the authority through interventions. The symmetric mean reversion property of the fundamental was used to model intervention conducted by central banks in a target-zone regime [54,55]. However, the mean-reverting forces contributed by the two terms in the proposed model are not symmetric. The restoring force (weakening the domestic currency) given by $A_{-1}/\nu$ is in general stronger than that given by $A_{1}\nu$ (strengthening the domestic currency). The asymmetric mean-reverting fundamental dynamics is a reasonably realistic and analytically tractable way to capture a monetary authority’s intervention policy under capital inflows—more intensive intervention when the currency appreciates and the fundamental moves closer to the boundary than that when the currency depreciates. Regarding the fundamental dynamics, increasing the magnitudes of the parameters $A_{-1}$ or $A_{1}\;$enhance the mean-reverting force for the fundamental, such that the fundamental variable is well bounded from the boundary, i.e., reducing the probability of v breaching the origin. From an empirical point of view, the asymmetric mean-reverting property with a relatively large $A_{-1}$ is consistent with the SNB’s intervention policy under which most of its interventions were conducted to counteract the upward pressure of the Swiss franc, including its exchange rate management during the coronavirus pandemic in early 2020. Section 4 discusses the relationship between the accumulation of foreign reserves due to interventions conducted by the SNB and the exchange rate dynamics derived from the model.

## 3. Results

### 3.1. Exchange Rate Solution

By the power series method, the optimal approximate solution of Equation (3) with the boundary conditions specified by Equations (4) and (5) is:(8)$$s\left(\nu \right)\approx -\frac{\epsilon m}{\alpha \left(\sigma_{\nu }^{2}+2A_{-1}\right)}\nu^{2}=\epsilon B_{0}\nu^{2}$$
where ϵ is a positive parameter determined by minimising the total error between the approximate solution and the power series solution.

Figure 2 plots the relationship between the exchange rate *S* in the original exchange rate measure and the fundamental *ν* expressed in Equation (8) based on estimations using the market data of the Swiss franc against the euro exchange rate. Substituting the expression of Equation (8) into Equation (1) yields $s=m+\sqrt{1/\left|\varepsilon B_{0}\right|}\sqrt{\left|s\right|}+\alpha E\left[ds\right]/dt$. The absolute numbers are taken as *s* is negative. E[ds]/dt is approximated by a six-month moving average of ds/dt. From the time series of s, we construct the time series of both $\sqrt{\left|s\right|}$ and E[ds]/dt. The parameters $m,\;\sqrt{1/\left|\varepsilon B_{0}\right|}$, and α can be determined by the least square regression with a three-year rolling window.

The figure shows that changes in the exchange rate flatten with changes in the fundamental at the boundary of EUR/CHF = 1.15. This means that the exchange rate could only marginally move away from the boundary due to interventions, even though the fundamental changes materially. When the exchange rate moves towards its boundary due to a positive demand shock in the fundamental, there is a counteracting tendency of a mean reversion back to the mean level, which acts as a stabilising force as shown in Equation (7) to limit further appreciation in the exchange rate. Based on the model, the exchange rate could move from C to C′ or C″ with changes in the fundamental, where the paths depend on the coefficient $\epsilon B_{0}$ in Equation (8). The coefficient $\epsilon B_{0}$ represents the state of the economy of the currency, including the money supply (*m*), parameters ($A_{-1}$) of the asymmetric fundamental shock, and sensitivity (*α*) of the exchange rate to its expected rate of change.

Given that UIP continually holds in the model of Equation (1), we have:(9)$$r\left(t\right)-r^{*}\left(t\right)=\frac{E\left[ds\left(t\right)\right]}{dt}$$
where *r*(*t*) and *r**(*t*) are the domestic and foreign interest rates, respectively, with tenor *t*. When the domestic interest rate *r* is at the ZLB, the appreciation expectation of the currency cannot be effectively reflected by lowering the interest rate via UIP and the interest rates become less responsive to the expectation of future exchange rate change. The corresponding effect of E[ds(t)]/dt on the exchange rate expressed in Equation (1) is therefore limited, suggesting a smaller value of semi-elasticity *α* of the exchange rate with respect to its expected rate of change. According to Equation (8), a smaller *α* makes *B*_0_ larger, such that the exchange rate will follow the path from C to C′ (with larger *B*_0_) instead of C″ in Figure 2. This demonstrates that the exchange rate is more sensitive to changes in the fundamental as illustrated from the figure when the domestic interest rate is at the ZLB. An extremely low interest rate environment may therefore trigger more interventions conducted by the monetary authority to prevent the appreciation of the currency and increase the money supply *m*.

To demonstrate the exchange rate dynamics, it is convenient to concentrate on the magnitude of *s* and introduce the new variablex≡−s, i.e., 0≤x<∞ with x=0 corresponding to the boundary. By applying Ito’s lemma to the Rayleigh process for the fundamentals ν of Equation (2) with Equations (7) and (8), *x* is shown to follow a mean-reverting square-root (MRSR) process:(10)$$dx=\kappa \left(\theta -x\right)dt+\sigma_{x}\sqrt{x}dZ$$
where
(11)$$\kappa =2\left|A_{1}\right|,\;\theta =\varepsilon \left|\frac{B_{0}}{A_{1}}\right|\left(A_{-1}+\frac{1}{2}\sigma_{\nu }^{2}\right),$$
(12)$$\sigma_{x}=2\sigma_{\nu }\sqrt{\left|B_{0}\right|}\;.$$In Equation (10), *κ* determines the speed of the mean-reverting drift towards the long-term mean θ.

According to Feller’s classification of boundary points, it can be inferred that there is a non-attractive natural boundary at infinity and the one at the origin is a boundary of no probability leakage (a condition determines whether *x* can fall below zero) for $\left(\frac{\sigma_{x}^{2}}{4\kappa \theta }\right)<1$ in Equation (10), and it is not otherwise [56]. The no-leakage condition ensures the exchange rate will not breach the origin (the boundary) and there is no large revaluation of the currency. If the no-leakage condition does not hold at the boundary, the smooth-pasting condition of Equation (5) may break down in the model and foreign exchange interventions fail with substantial appreciation of the currency. Therefore, the exchange rate is quasi-bounded at the origin.

Conventionally, a transition between the floating-rate and target-zone regimes seems to be abrupt in terms of exchange rate dynamics. The analysis here shows that when interventions are present in both regimes, the transition between the two regimes via appropriate normalisation of the log exchange rates causes changes in the level of exchange rate volatility but not the dynamical structure, suggesting that choices between the floating-rate and target-zone regimes seem to have little consequence in this regard. Figure 4 illustrates that the log exchange rate of the Swiss franc in *x* normalised by the ceiling at 1/1.2 EUR/CHF in the target-zone regime (September 2011–January 2015) has smaller variations than that of the log exchange rate normalised by the moving boundary, but they have similar dynamical movements.

### 3.2. Interest Rate Differential at ZLB

With and without revaluation risk, the foreign exchange risk premium is assumed to be zero [57]. Based on UIP in the model with Equation (9), the interest rate differential $\delta \left(x,t\right)=\left(r-r^{*}\right)$ is [17]:(13)$$\delta \left(x,t\right)=\left(\theta -x\right)\left[\frac{1-\exp\left(-\kappa t\right)}{t}\right]$$

The effect of UIP in the exchange rate model is reflected in the gap between the spot and mean exchange rates in the exchange rate dynamics (i.e., θ−x), which determines the interest rate differential depending on the expectation on the exchange rate. Equation (13) suggests that the appreciation expectation on the domestic currency is reflected from the exchange rate mean level *θ* being stronger than the spot rate *x* momentarily (i.e., x>θ), such that the interest rate differential δ(x,t) is negative (i.e., *r* < *r**). If the appreciation expectation is extremely strong with θ≈0, the interest rate differential is $\delta \left(x,t\right)\approx -x\left[\frac{1-\exp\left(-\kappa t\right)}{t}\right]$. The relative position of the spot *x* and mean *θ* exchange rates determines the sign and magnitude of the interest rate differential. The stronger appreciation expectation suggests a larger distance between *x* and *θ* and a wider negative δ(x,t). If the home country is not at the ZLB, a lower domestic interest rate can offset the currency expectation. It is therefore not necessary for the monetary authority to intervene in the foreign exchange market and prevent currency appreciation.

When the domestic interest rate is at the ZLB, the currency appreciation expectation cannot be offset by lowering the interest rate, such that the negative interest rate differential cannot be widened further. Due to such constraint, the corresponding distance between *x* and *θ* in the exchange rate dynamics cannot be widened either. As a result of the appreciation expectation, both *x* and *θ* will strengthen accordingly. To counteract the upward pressure on the currency, the monetary authority needs to intervene and increase its foreign exchange reserves. The intervention aims to reduce the appreciation pressure by preventing the exchange rate *x* from moving towards the strong-side boundary and weakening the mean level *θ*. Based on the expression of the interest rate differential in Equation (13) derived from the model and UIP, the corresponding exchange rate dynamics captures the effect of the domestic interest rate at the ZLB. The linkage between foreign reverse accumulation and the exchange rate dynamics is investigated empirically using the Swiss franc exchange rate in Section 4.

### 3.3. Exchange Rate Distribution

The probability density function (PDF) of *x* under the MRSR process is given by:(14)$$\begin{array}{cc} G\left(x,t;x^{\prime },t^{\prime }\right)=\frac{2}{\sigma_{x}^{2}C_{1}\left(t-t^{\prime }\right)}{\left(\frac{x}{x^{\prime }}\right)}^{\omega /2}\exp\left[-\frac{\omega +2}{2}C_{2}\left(t-t^{\prime }\right)\right]\times & \\ \exp\left\{-\frac{2x^{\prime }+2x\exp\left[-C_{2}\left(t-t^{\prime }\right)\right]}{\sigma_{x}^{2}C_{1}\left(t-t^{\prime }\right)}\right\}\times & ,\\ I_{\omega }\left\{\frac{4x^{1/2}x\prime^{1/2}\exp\left[-C_{2}\left(t-t^{\prime }\right)/2\right]}{\sigma_{x}^{2}C_{1}\left(t-t^{\prime }\right)}\right\} & \end{array}$$
where $\omega =2\kappa \theta /\sigma_{x}^{2}-1$, $C_{1}\left(\tau \right)=\left[\exp\left(\kappa \tau \right)-1\right]/\kappa$, and $C_{2}\left(\tau \right)=-\kappa \tau$, $I_{\omega }$ is the modified Bessel function of the first kind of order ω. The associated asymptotic PDF will eventually approach the steady-state exchange rate distribution, which is:(15)$$K\left(x,t\to \infty ,x^{\prime },t^{\prime }\right)=\frac{2x^{\omega }}{\Gamma \left(\omega +1\right)}{\left(\frac{2\kappa }{\sigma_{x}^{2}}\right)}^{\omega +1}\exp\left(-\frac{2\kappa }{\sigma_{x}^{2}}x\right),$$
where Γ is the gamma function. Given the PDF in Equation (14), the parameters of the MRSR process for the exchange rate dynamics are calibrated in Section 3.4 using market exchange rate data of the Swiss franc. The exchange rate distribution given by Equation (15) is skewed. It is noted that the multivariate skew normal distribution can generate similar skewed distributions by adjusting the shape parameter [58]. The skewness in Equation (15) is, however, determined by the location of the long-term mean θ relative to the boundary at zero and the mean-reverting parameter *κ*.

Using the Swiss franc’s exchange rate against the euro (EUR/CHF) for illustration, Figure 3 shows the steady-state exchange rate distributions in the original exchange rate *S* based on Equation (15) with three values of the long-term mean θ of 0.05, 0.15, and 0.25 (*θ_S_* = 1.094, 0.9898, 0.8956 in *S*): the smaller θ is closer to the boundary at *x* = 0 (*S_boundary_* = 1.15). We use the model parameters for *σ_x_* = 0.01 (Panel A) and 0.03 (Panel B), and *κ* = 0.02, which are consistent with the estimations in Section 4. The distributions with *θ_S_* = 1.094 and 0.9898 have their peaks at the right, showing the PDF decays slower than a Gaussian distribution at the left, suggesting the fat-tails effect with the probability of outlier negative returns. The exchange rate distributions have fatter tails with the mean *θ_S_* closer to the boundary, demonstrating that the probability of outlier negative returns becomes more significant for the currency expected to appreciate closer to the boundary in the near term. This demonstrates that the monetary authority would intervene in the foreign exchange market according to the policy specified in the fundamental dynamics in Equation (7) and push the exchange rate lower, such that the distributions show outlier negative returns. The intervention is more intensive with the mean *θ_S_* closer to the boundary, as the monetary authority wants to prevent the currency appreciating further. Comparison between Panels A and B, where *σ_x_* increases from 0.01 to 0.03, shows the left tails of the distributions become much fatter, and their left-skewness is sensitive to an increase in the exchange rate volatility. This suggests that the higher exchange rate volatility increases the likelihood of negative excess returns due to interventions.

### 3.4. Model Calibrations for Swiss Franc

We calibrate the MRSR process for the exchange rate dynamics using the Swiss franc against the euro under the floating-rate regime during 15 January 2015–26 March 2020. A moving boundary is specified as a moving average *S_A_* scaled by a parameter $\eta_{U}$, such that $\eta_{U}S_{A}\left(t\right)$ forms the boundary for the exchange rate movement. The parameter $\eta_{U}$ reveals how far the SNB tolerates an appreciation of the Swiss franc. The parameter $\eta_{U}$ in Equation (6) is the upper boundary set at 1.25 and$\;S_{A}\left(t\right)$ is defined as a six-month moving average. Therefore, the boundary, which is set at about 25% above $S_{A}\left(t\right)$, can be considered as “large revaluation” given that the observed exchange rate appreciated by about 15% in a few days after removal of the strong-side ceiling at 0.8333 EUR/CHF of the target-zone regime on 14 January 2015. Figure 1 shows the EUR/CHF exchange rate in *S* and the corresponding moving and fixed boundaries for the floating-rate and target-zone regimes, respectively. The distances between the exchange rate and moving boundary were large and only narrowed in short periods before and after the target-zone regime when the exchange rate surged. The normalised log exchange rate in *x* is shown in the upper panel of Figure 4. For comparison, the exchange rate normalised by the strong-side ceiling during the target-zone regime is shown to have similar movements as those of the exchange rate normalised by the moving boundary but with smaller magnitude, indicating that the level of the boundary does not change the dynamical structure of the normalised exchange rate much and is simply a scaling issue.

The maximum likelihood estimation using daily data with a rolling 3-year window is used to estimate the model parameters in Equation (10) of the exchange rate dynamics based on the log-likelihood function that is constructed by the analytical PDF of Equation (14). The estimated daily volatility $\sigma_{x}$ shown in Panel A of Figure 5 ranges between 0.006 and 0.007. The corresponding *z*-statistic is much higher than 1.96 (i.e., at the 5% significance level), indicating that the estimated $\sigma_{x}$ is highly significant. This demonstrates that the estimation of the square-root process is consistently robust for the estimation period. While the changes of $\sigma_{x}$ over time are within a relatively narrow range, the next section shows that its variation is related to the accumulation of the foreign reserves through the interventions conducted by the SNB.

Panel B shows that the estimates of the drift term *κ* are significant in terms of the *z*-statistic and range between 0.01 and 0.025. Similar to Panel B, Panel C demonstrates that the estimated mean *θ* is significant at the level of about 0.22 and time varying. The estimations for *κ* and *θ* indicate that the mean reversion is significantly present in the exchange rate dynamics as specified in the model. The explicit calibration of the model parameters in Equation (10) shown in Figure 5 confirms that the CHF exchange rate dynamics follows the MRSR process with the corresponding normalisation of the log exchange rate.

The lower panel of Figure 4 illustrates that the signal of the probability-leakage ratio $\left(\frac{\sigma_{x}^{2}}{4\kappa \theta }\right)$ is close to zero during most of the estimation period, indicating no probability leakage. For illustrative purposes, we extend the estimation before 15 January 2015. The probability-leakage ratio before 15 January 2015 is shown to have jumps in the short periods before and after the target-zone period, when the CHF appreciated sharply against the euro and the exchange rate is very close to the moving boundary as shown in Figure 1.

## 4. Discussion

A monetary authority pursuing an exchange rate policy is inconsistent with interest rate parity because of a binding ZLB constraint [2]. To prevent currency appreciation, the monetary authority needs to absorb capital inflows by accumulating foreign reserves. Much larger changes in foreign reserves are required to equilibrate currency markets when interest rates are zero, as an expectation of exchange rate appreciation will cause foreign reserves to swell [59].

The estimation period of the model parameters in the previous section covers the period when the nominal interest rates are at zero or even below zero at home (Switzerland) and abroad (the euro area). The Swiss franc which is considered as a safe-haven currency will be attractive to foreign investors as the expected future appreciation of the Swiss franc is not offset by lower domestic interest rate. To prevent further appreciation, the SNB need to absorb the capital inflows through foreign exchange interventions, such that its foreign reserves will increase accordingly. These interventions also affect the exchange rate dynamics. Therefore, the estimated model parameters of the dynamics, in particular the volatility $\sigma_{x}$, and the distance (Δ*θ_S_*) between the mean level *θ_S_* of the exchange rate and the moving boundary $\eta_{U}S_{A}\left(t\right)$ in the original exchange rate *S* are expected to be related to the foreign reserves. The original exchange rate *S* is used because one of the objectives of the SNB’s interventions is to target the level of the exchange rate in *S,* not the normalised exchange rate in *x,* in order to counteract the upward pressure. In the model, the mean level *θ_S_* reflects the expectation of the exchange rate as illustrated by Equations (9) and (13) relating to the interest rate differential. The reserves reported in the Swiss franc comprise about 46% in the euro, 28% in the US dollar, and the rest in the Japanese yen, sterling, and others. Since only monthly information of the reserves is available, we use the month-end data for the estimations.

To investigate the relationship between the foreign reserves and exchange rate dynamics, we postulate that there is a long-run equilibrium relationship between {$\sigma_{x}$, Δ*θ_S_*} and the foreign reserves. The short-run dynamics represented as a dynamical error-correction model is given by:(16)$$\Delta y_{t}=a_{10}+\gamma \left(y_{t-1}-\beta_{y}\ln{\left(Reserves\right)}_{t-1}\right)+\sum_{k}b_{1k}\Delta y_{t-k}+\sum_{k}c_{1k}\Delta \ln{\left(Reserves\right)}_{t-k}$$
where γ is less than zero. $y_{t}$ representing ln{$\sigma_{x}$, Δ*θ_S_*} will change in response to the reserves and to the previous period’s gap from the long-run equilibrium (i.e., $y_{t-1}-\beta_{y}\ln{\left(Reserves\right)}_{t-1}$). The parameter γ is the speed of adjustment. In absolute terms, the larger γ is, the greater the response of $y_{t}$ to the previous period’s gap from the long-run equilibrium. If γ is equal to zero, the long-run equilibrium relationship does not appear and the model is not an error-correction one or cointegrated. Therefore, for a meaningful cointegration and error-correction model, the speed of adjustment γ must be non-zero.

The estimation is conducted using month-end data for {$\sigma_{x}$, Δ*θ_S_*} and the foreign reserves in the period between January 2018 and February 2020. Table 1 provides the Augmented Dickey–Fuller (ADF) and Phillips–Perron test results for {$\sigma_{x}$, Δ*θ_S_*} and (Reserves) in levels and changes. It fails to reject at the 10% level the presence of a unit root for the variables in levels. However, the test for the first differences is significant at the 1% level. Therefore, the changes are stationary. This suggests that the variables considered are all I(1), i.e., integrated of the same order (1), which satisfies the requirement for the variables to be cointegrated.

To test the cointegration between ln(Reserves) and ln{$\sigma_{x}$, Δ*θ_S_*}, we use the Engle–Granger single-equation test [60], which is regarded as an easy and super-consistent method of estimation. It determines whether the residuals of the linear combination among the cointegrated variables estimated from the ordinary least squares method are stationary. Table 2 reports the cointegration tests between (Reserves) and {$\sigma_{x}$, Δ*θ_S_*}. The lag length is determined by the Schwartz criterion. The results are significant at the 1% or 5% levels. Thus, we reject the null hypothesis that (Reserves) and {$\sigma_{x}$, Δ*θ_S_*} are not cointegrated in favour of the alternative hypothesis that there is at least one cointegrating vector.

Table 3 reports the estimated cointegrating vectors. The coefficients *β* for {$\sigma_{x}$, Δ*θ_S_*} are significant at the 1% level. $\sigma_{x}$ is negatively and Δ*θ_S_* is positively related to (Reserves), suggesting that the accumulation of the foreign reserves stabilizes lises the exchange rate and weakens the long-term mean of the rate. The result supports the mechanism, in which a monetary authority needs to absorb capital inflows to dampen the currency appreciation by accumulating foreign reserves. Intuitively, it means that when capital inflows push the exchange rate to appreciate towards its “strong-side” boundary, the interventions of selling the Swiss franc to the market conducted by the SNB could alter the exchange rate expectation. The mean level of the exchange rate depreciates and moves away from the “strong-side” boundary $\eta_{U}S_{A}\left(t\right)$, i.e., an increase in Δ*θ_S_*. Under the mean-reverting force in the dynamics, the exchange rate is expected to move towards the long-term mean *θ_S_* and the currency depreciates accordingly. The results illustrate that the foreign exchange interventions conducted by the SNB weakened the Swiss franc, which is consistent with the objectives of its interventions to counteract the upward pressure on its currency. In addition, the SNB’s interventions generally decreased exchange rate volatility. The result is consistent with an objective of foreign exchange intervention to stabilise the exchange rate of the currency by reducing its volatility.

Finally, Table 4 reports the estimates of the short-run dynamics. The speeds of adjustment γ are negative at the 5% or 10% significance levels and smaller than 1 in absolute value. This suggests that the error correction specification is valid and there is a self-restoring force to close the gap between {$\sigma_{x}$, Δ*θ_S_*} and (Reserves), and subsequently to restore the long-run equilibrium.

The empirical results demonstrate that the SNB’s interventions, which increased its foreign reserves, had two different effects during 2015–2020. First, the interventions stabilised the exchange rate by reducing volatility. Second, the interventions weakened the long-run mean level in the exchange rate dynamics, which reflects the diminished appreciation expectation. This lowered the attractiveness of Swiss franc investments and thus counteracted the upward pressure on the currency. The results suggest that the SBN in general achieved its policy objectives of interventions, namely, diminishing appreciation pressure of the currency’s exchange rate level (first moment) and stabilising the exchange rate volatility (second moment).

## 5. Conclusions

The appreciation pressure on a currency cannot be offset by lower domestic interest rates because of a blinding zero lower bound (ZLB) constraint. A monetary authority needs to intervene in the foreign exchange market and accumulate foreign reserves to counteract upward pressure on its currency. This paper develops an exchange rate model that incorporates with intervention policy by specifying the fundamental dynamics following the Rayleigh process with the smooth-pasting condition at a boundary. The exchange rate dynamics derived from the model follows the mean-reverting square-root process (MRSR), and its associated left-skewed exchange rate distribution shows negative excess returns due to a monetary authority’s interventions to dampen currency appreciation. The probability-leakage condition for the exchange rate breaching the boundary forms a measure of uncertainty about degree of effectiveness of interventions. The interest rate differential derived from the solution of the model is determined by the difference between the spot and mean level in the exchange rate dynamics. Because of the ZLB constraint, the interest rate differential cannot be widened by lowering the domestic interest rate to offset upward pressure on the currency. The corresponding difference between the spot and mean exchange rates in the exchange rate dynamics cannot be widened either. The exchange rate is therefore more sensitive to the change in the fundamental, suggesting more intensive interventions required to counteract the upward pressure.

The empirical results demonstrate that the model can be calibrated by the Swiss franc exchange rate data under the floating-rate regime during January 2015–February 2020 when the domestic interest rate was at the ZLB. The results about the relationship between the exchange rate dynamics and the SBN’s foreign reserves accumulated due to interventions suggest that the SBN in general achieved its policy objectives of interventions. First, the interventions diminished appreciation pressure of the currency’s exchange rate level by weakening the mean level in the exchange rate dynamics. Second, they stabilised the exchange rate volatility.

An implication of this study is that, by incorporating a monetary authority’s intervention policy, the exchange rate dynamics under both the target-zone and floating-rate regimes follow the same process via appropriate normalisations of the log exchange rates. The transition between the two regimes causes changes in the level of exchange rate volatility but not its dynamical structure, suggesting that transitions between the floating-rate and target-zone regimes do not seem to have material consequence in this regard.

The exchange rate model assumes a continuous-time representation and that there is an equilibrium that is stable. These assumptions have limitations given that the exchange rate may not be in stable equilibrium under economic shocks [21,22]. The exchange rate is not a conserved quantity and could have chaotic fluctuations [28]. Therefore, a discrete-time exchange rate model is for future research.

## Figures and Tables

**Figure 1 entropy-24-00888-f001:**
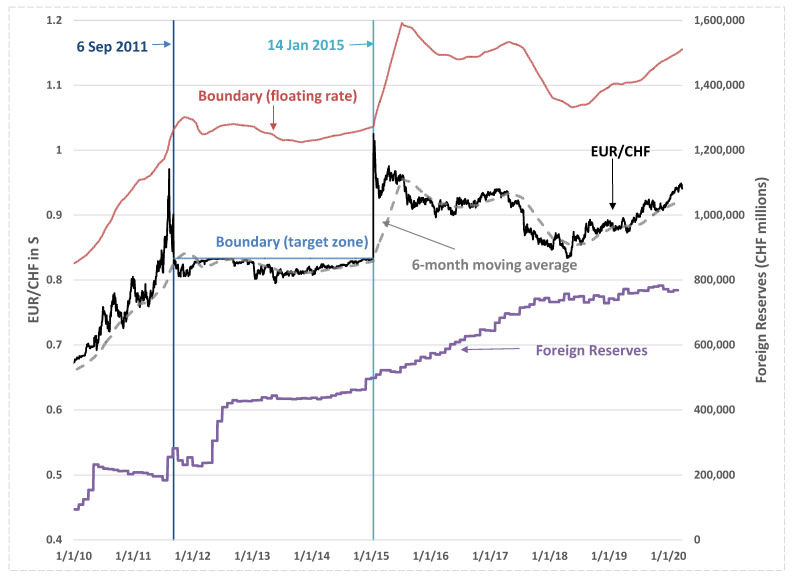
Swiss franc (CHF) against euro (EUR) exchange rate (*S*), its 6-month moving average, floating-rate boundary, target-zone boundary (6 September 2011–14 January 2015), and foreign reserves in Swiss franc.

**Figure 2 entropy-24-00888-f002:**
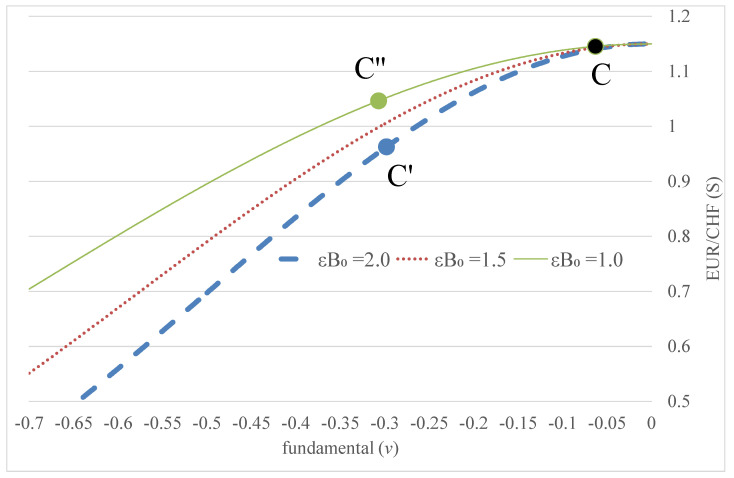
Relationship between Swiss franc (EUR/CHF) exchange rate (*S*) and fundamental (*ν*) based on Equation (10) with $\epsilon B_{0}$ = 1, 1.5 and 2.

**Figure 3 entropy-24-00888-f003:**
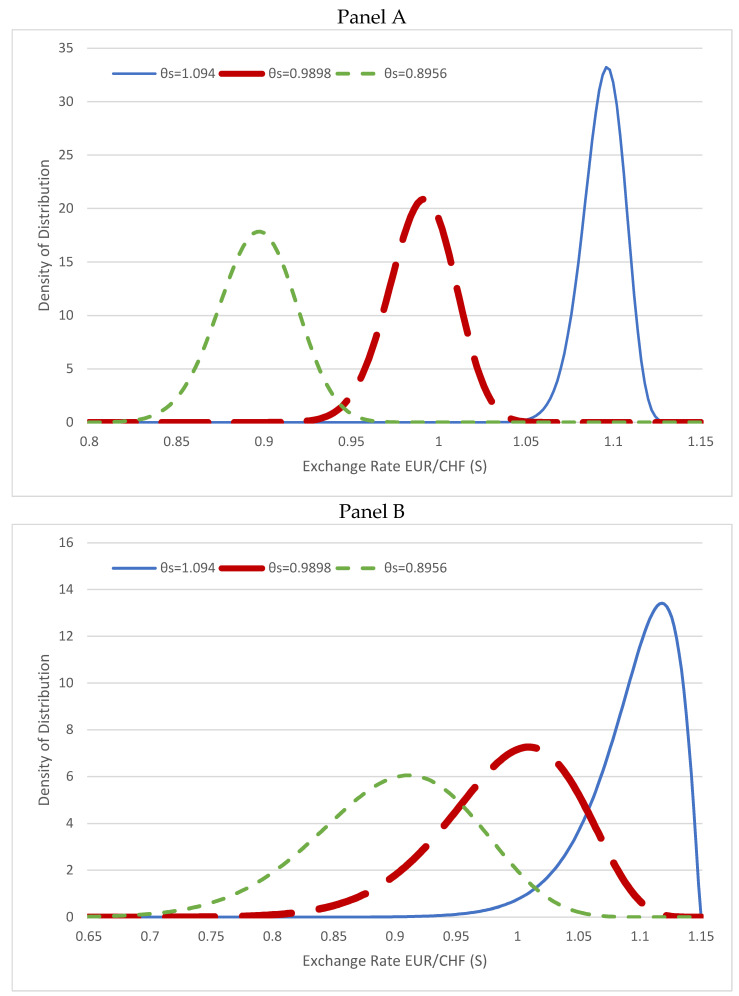
Exchange rate distributions of EUR/CHF (*S*) based on Equation (15) with *κ* = 0.02, *θ* = 0.05, 0.15, 0.25 (*θ_S_* = 1.094, 0.9898, 0.8956), *σ_x_* = 0.01 (Panel **A**), and *σ_x_* = 0.03 (Panel **B**).

**Figure 4 entropy-24-00888-f004:**
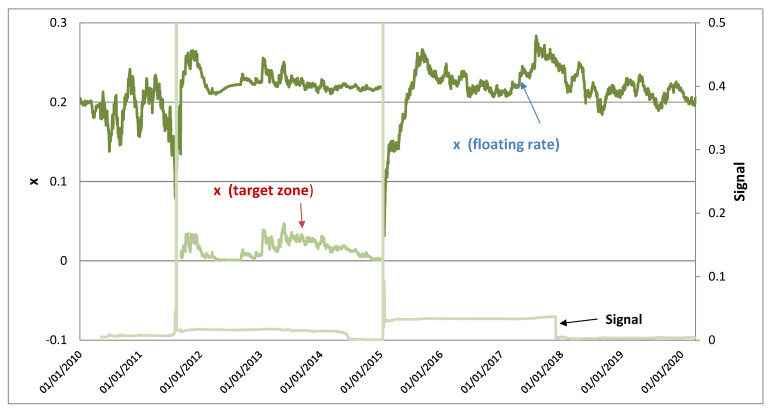
Swiss franc (EUR/CHF) exchange rate (*x*) under floating-rate and target-zone regimes respectively and probability-leakage ratio (signal)l)$\;\left(\frac{\sigma_{x}^{2}}{4\kappa \theta }\right)$.

**Figure 5 entropy-24-00888-f005:**
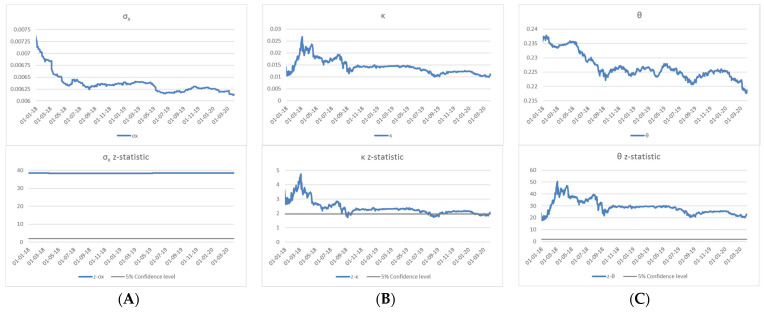
Estimated model parameters of EUR/CHF exchange rate (*x*) with 3-year rolling window: (**A**) estimates and *z*-statistic of $\sigma_{x}$, (**B**) estimates and *z*-statistic of *κ*, and (**C**) estimates and *z*-statistic of *θ*.

**Table 1 entropy-24-00888-t001:** Descriptive statistics, ADF, and Phillips–Perron tests.

	ln(*σ_x_*)		ln(Δ*θ_S_*)		ln(Reserves)	
	Monthly Last		Monthly Last		Monthly	
	Level	Change	Level	Change	Level	Change
Mean	−5.06	0.00	−1.50	0.00	13.53	0.002
Median	−5.06	0.00	−1.50	0.00	13.53	0.003
Maximum	−4.98	0.01	−1.47	0.02	13.57	0.026
Minimum	−5.09	−0.05	−1.54	−0.01	13.50	−0.027
Std. Dev.	0.03	0.01	0.02	0.01	0.02	0.015
Skewness	1.79	−1.73	−0.03	0.33	0.05	−0.402
Kurtosis	6.19	6.37	2.61	2.27	1.82	2.349
ADF test statistics	1.67	−3.394 ***	−0.28	−4.550 ***	−3.21	−6.017 ***
Phillips–Perron test statistics	1.67	−3.216 ***	−0.27	−4.551 ***	−3.19	−6.570 ***
Correlation (between ln(Reserves) and *y*)	−0.652	-	0.605	-	-	-
Observations	26	25	26	25	26	25

Notes: *** indicates significance at a level of 1%. Both tests check the null hypothesis of unit root existence in the time series, assuming nonzero mean in the test equation.

**Table 2 entropy-24-00888-t002:** Test of cointegration (Euler-Granger).

Dependent Variable	ln(*σ_x_*)	ln(Δ*θ_S_*)
	Monthly	Monthly
ADF test statistic	−3.64 ***	−2.09 **
Phillips–Perron test statistic	−3.66 ***	−2.26 **

Notes: *** and ** indicate significance at levels of 1% and 5%, respectively. The cointegration test uses the Augmented Dickey–Fuller and Phillips–Perron tests to check the null hypothesis that the residuals of the regression of ln(Reserves) and the parameters from the MLE calibration with the 3-years rolling window are non-stationary, assuming zero mean in the test equation. The critical value of the test is obtained from MacKinnon (1996) [61].

**Table 3 entropy-24-00888-t003:** Estimates of the long-run part (*β*) of cointegrating vectors.

Dependent Variable	ln(*σ_x_*)	ln(Δ*θ_S_*)
	Monthly	Monthly
ln(Reserves)	−0.83 ***	0.49 ***
Constant	6.17 *	−8.11 ***

Notes: *** and * indicate significance at levels of 1% and 10% respectively.

**Table 4 entropy-24-00888-t004:** Estimation results of short-run dynamics (*γ*).

Dependent Variable	ln(*σ_x_*)	ln(Δ*θ_S_*)
	Monthly	Monthly
ln(Reserves)		
Speed of adjustment	−0.34 *	−0.55 **
Lag length	4	4

Notes: ** and * indicate significance at levels of 5% and 10% respectively.

## Data Availability

The datasets generated during and analysed during the current study are available from Bloomberg and the Swiss National Bank at https://data.snb.ch/en/topics/snb#!/cube/snbimfra, (accessed on 1 April 2020).

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
