# Peer review of "Modelling Foreign Exchange Interventions under Rayleigh Process: Applications to Swiss Franc Exchange Rate Dynamics"

_entropy, 2022, doi:10.3390/e24070888_

Round 1
Reviewer 1 Report
The paper is an interesting paper for monetary policy. This paper modeled the foreign exchange intervention policy following the Rayleigh process derived from the standard flexible-price monetary frame work. This is a good approach but the authors need to compare the proposed process with other distribution function, skewed t-distribution developed by Azzalini and Capitanio (2003).
Azzalini, Adelchi, and Antonella Capitanio. 2003. “Distributions Generated by Perturbation of Symmetry with Emphasis on a Multivariate Skew t-Distribution.” Journal of the Royal Statistical Society: Series B (Statistical Methodology) 65 (2): 367–89.
Minor comment:
There is typo mistake (..) such as
2.1. . Basic exchange rate model and boundary conditions
The authors need to check the manuscript carefully when they revise the paper.
If the authors follow my suggestion, I will be happy to recommend this paper for possible publication from entropy.
Author Response
We appreciate the comments and suggestions that greatly improve the presentation. The responses are as follows:
- This is a good approach but the authors need to compare the proposed process with other distribution function, skewed t-distribution developed by Azzalini and Capitanio (2003).
Response:
In the section “Results” 3.3, it discusses that the multivariate skew normal distribution can generate similar skewed distributions by adjusting the shape parameter [Azzalini and Capitanio (2003)]. The skewness in Eq.(15) is however determined by the location of the long-term mean θ relative to the boundary at zero and the mean-reverting parameter k.
- Minor comment: The authors need to check the manuscript carefully when they revise the paper.
Response:
They are corrected. And some of them will be fixed by the Journal’s editing.
Reviewer 2 Report
This paper discusses exchange rate mechanisms quantitatively. The special point are interest rates around zero where lowering interest rates is impossible.
The authors give a sound discussion based on random processes. They show explicit results which represents reality well.
However, the authors should add some critical remark on the validity of the models they use. It will improve the introduction and discussion. The referee sees the following points:
· Exchange rates are not conserved quantities. So there is not necessarily an equilibrium (like in Brownian motion) where the exchange rate fluctuates around
· Even if there is an equilibrium, it is no necessarily stable.
· Fluctuations in finance are generally not random they are chaotic.
Some more details are given in the comments in the original manuscript which I will include. There some minor points are also highlighted such as bold and/or bigger symbols in the text.

Author Response
We appreciate the comments and suggestions that greatly improve the presentation. The responses are as follows:
- Exchange rates are not conserved quantities. So there is not necessarily an equilibrium (like in Brownian motion) where the exchange rate fluctuates around Even if there is an equilibrium, it is no necessarily stable. Fluctuations in finance are generally not random they are chaotic.
Response: These limitations of the model are discussed in the “Introduction” and “Conclusion” with the suggested references of [22, 23, 28].
- Some more details are given in the comments in the original manuscript which I will include. There some minor points are also highlighted such as bold and/or bigger symbols in the text.
Response:
- As every model, the target zone model [35] has a range of validity. How is this range? Is it fulfilled here?
It is specified after Eq. (1).
- Please note that the exchange rate is set by the market. It does not exist between tradings. A continuous interpolation is not necessarily correct as the exchange rate is not a conserved quantity, see e.g. [24a].
This limitation of the model is mentioned in the first paragraph of the section “Method” and in the conclusion.
- As "E" is the expectation operator, do you mean "d/dt E[s(t)]" or "E[ds(t)/dt]" in Eq. (1)? As long as E is linear, it is of course irrelevant.
E is linear. It means "d/dt E[s(t)]".
- Is a time series of Sqrt[Abs[s]] possible?
The absolute numbers are taken given that s is defined as negative in the model.
- In this discussion the limitations of the models used here should be addressed. Besides [21a], [21b], and [24a] there maybe more. From it one may also get an idea for future work.
The limitations of the model are discussed in the “Conclusion” with the suggested references of [22, 23, 28].
- There some minor points are also highlighted such as bold and/or bigger symbols in the text.
They are corrected. And some of them will be fixed by the Journal’s editing.